# Deep Multi-view Graph Clustering via Attribute-aware Bidirectional Structural Refinement and Pseudo-label Guided Multi-level Fusion

**Youqing Wang** [1]   **Tianxiang Zhao** [1]   **Mengyuan Xin** [1]   **Ye Su** [2]   **Jiapu Wang** [3]   **Tengfei Liu** [4]   **Junbin Gao** [5]
**Jipeng Guo** [1]

## Abstract

Deep multi-view graph clustering (DMGC) typically leverages graph neural networks for representation learning, but most existing methods excessively depend on local and static graph structures and only utilize simplistic cross-view fusion strategies. To this end, this paper proposes **A**ttribute-aware Bidirectional Structural Refinement (ABSR) and **P**seudo-label Guided Multi-level Fusion (PGMF) for DM**GC**, termed **APGC**. Specifically, ABSR selectively strengthens high-quality connections and suppresses semantically conflicting relationships, achieving bidirectional refinement of the graph structure based on attribute similarity. It incorporates global attribute semantics into the graph structure, thereby promoting the homophilic connections for discriminative graph representation learning. Guided by reliable pseudo-labels, PGMF achieves adaptive weighted fusion at both the node-level and the view-level, effectively balancing the differentiated contributions of multi-view information. Experiments on six homophilic and heterophilic datasets demonstrate the superior clustering performance of the proposed APGC method. The code of APGC could be available at https://github.com/TianxiangZhao0474/APGC.git.

## 1. Introduction

Graph data, characterized by its unique ability to simultaneously model node attributes and topological structures, plays an indispensable role in domains such as social networks (Huang et al., 2025; Wang et al., 2025c), citation analysis (Liu et al., 2024; Yang et al., 2024; Zhao et al., 2026a; Wu et al., 2026), and bioinformatics (Li et al., 2025; Chen et al., 2025b). In recent years, Graph Neural Networks (GNNs) have achieved significant progress in tasks like graph clustering (Lu et al., 2025; Wang et al., 2025a; Zhao et al., 2026b), traffic prediction (Guo et al., 2020), and knowledge graph analysis (Liang et al., 2024), owing to their powerful representation learning capabilities in non-Euclidean spaces. However, as real-world application scenarios become increasingly complex, a single view (Wang et al., 2025b) is often insufficient to fully capture the multidimensional associations between nodes. Consequently, Deep Multi-view Graph Clustering (DMGC) (Lin et al., 2025b; Chen et al., 2025c; Zhao et al., 2025; Guan et al., 2026) has emerged, aiming to achieve precise unsupervised node partitioning by integrating complementary information from multiple views.

Recently, GNNs-based DMGC methods are popular and have evolved from generative paradigms to contrastive ones. Early pioneers like One2Multi graph Autoencoder Clustering (O2MAC) (Fan et al., 2020) utilized graph autoencoders to reconstruct multi-view graph information, while Multi-view Attribute Graph Convolution Networks for clustering (MAGCN) (Cheng et al., 2021) further designed a dual-path encoding architecture to enhance intra-view and inter-view interactions. To improve the discriminative capability of embedding representations, methods such as contrastive Multi-View Representation Learning on Graphs (MVGRL) (Hassani & Khasahmadi, 2020) and Multi-view Contrastive Graph Clustering (MCGC) (Pan & Kang, 2021) introduced contrastive learning mechanisms, learning consistent representations by maximizing mutual information across views or scales. Furthermore, to address the heterophily issue in graph, recent works like towards self-supervised Learning on Graphs with Heterophily (HGRL) (Chen et al., 2022), Graph Representation learning method with Edge hEterophily discriminaTing (GREET) (Liu et al., 2023), and inCOMPlete muLti-view clustEring via conTrastivE pRediction (COMPLETER) (Lin et al., 2021) have attempted to employ high-pass filters or dual-prediction mechanism to preserve original feature patterns within non-homophilic structures.

---

[1]College of Information Science and Technology, Beijing University of Chemical Technology [2]University of the Chinese Academy of Sciences [3]Nanjing University of Science and Technology [4]Beijing University of Technology [5]The University of Sydney. Correspondence to: **Jipeng Guo** <guojipeng@buct.edu.cn>.

*Proceedings of the $43^{rd}$ International Conference on Machine Learning*, Seoul, South Korea. PMLR 306, 2026. Copyright 2026 by the author(s).

Despite these advancements, existing methods still face two crucial challenges when dealing with graph that possesses both complex attributes and multi-view heterophilic characteristics. First, the over-reliance on static local topology and the homophily assumption limits their performance in heterophilic scenarios, making them difficult to balance the node-itself and neighboring information. Most DMGC methods perform message passing directly based on the raw adjacency matrix, implicitly assuming strong homophily. However, heterophilic structures are ubiquitous in real-world data. As noted by (Tang et al., 2022) and (Chen et al., 2022), a delicate trade-off exists in heterophilic graphs where excessive aggregation of neighborhood information introduces structural noise. This issue may result in over-smoothed representations that may underperform even simple Multilayer Perceptrons (MLPs). On the other hand, relying solely on node attributes to avoid structural noise forfeits the contextual advantages of GNNs. Although methods like Decoupled Self-Supervised Learning for graphs (DSSL) (Xiao et al., 2022) and multi-view graph clustering via Node-Guided Contrastive Encoding (NGCE) (Ren et al., 2025) attempt to mitigate this by decoupling heterophilic patterns or introducing node guidance, they lack a unified mechanism to dynamically refine the structure using global attribute semantics. Consequently, in extreme scenarios with very low homophily, these methods fail to effectively eliminate semantic conflict edges and thereby limit clustering performance. Second, existing multi-view fusion strategies lack node-level adaptivity and global consistency collaboration. Multi-view fusion is the core of DMGC, yet current strategies (Shen et al., 2024) mostly adopt static or view-level weighting schemes. This overlooks the fact that the informativeness of a view varies significantly at node granularity, meaning that the contribution across different nodes may differ drastically in same view. Such coarse fusion methods, lacking node-level adaptive perception, struggle to strike a balance between preserving node personality and maintaining global cross-view consistency, resulting in consensus embeddings that are deficient in representation discriminability and stability.

To this end, this paper proposes Attribute-aware Bidirectional Structural Refinement (ABSR) and Pseudo-label Guided Multi-level Fusion (PGMF) for DMGC, termed APGC. Specifically, the ABSR module utilizes augmented attribute representations to perform dynamic bidirectional refinement on the topological structure. By precisely reinforcing high-quality connections and eliminating semantic conflict edges, this module deeply integrates global attribute semantics into the local topology, effectively breaking the representation learning bottleneck of traditional methods in heterophilic scenarios. Further, the PGMF module designs a pseudo-label guided multi-level adaptive fusion mechanism, synergizing contributions from different views at both node

and view granularities. Coupled with a contrastive learning strategy that balances node consistency and view semantic alignment, PGMF achieves efficient integration and deep alignment of cross-view information. Through the synergistic optimization of ABSR and PGMF, APGC significantly enhances clustering performance.

For clarity, the main contributions of this study are summarized as follows:

- By leveraging global attribute semantics to dynamically reconstruct topological structure, ABSR precisely eliminates heterophilic noise and compensates for structural deficiencies, effectively overcoming homophily bottleneck inherent in representation learning.

- Through adaptive weighting and semantic alignment at both node and view granularities, PGMF resolves the issue of rigid fusion and significantly enhances the discriminability of the consensus representation.

- Extensive experiments on both homophilic and heterophilic datasets show that APGC achieves better performance over representative DMGC methods.

## 2. Model Formulation

This section presents a novel DMGC method, APGC, comprising two core modules ABSR and PGMF. The ABSR module utilizes augmented attribute representations to perform dynamic bidirectional structural refinement by deeply integrating global attribute semantics into local connections, generating semantically matching attribute embeddings and topology structures. Simultaneously, the PGMF module introduces a pseudo-label guided multi-level adaptive fusion mechanism that achieves precise integration of multi-view information through the synergistic optimization of multi-granularity weights to generate a highly discriminative consensus representation. The overall framework of APGC is illustrated in Figure 1.

### 2.1. Notations

For the multi-view graph $\mathcal{G} = \{\mathcal{V}, \mathcal{E}^1, \cdots, \mathcal{E}^V, \mathbf{X}\}$, $\mathcal{V} = \{v_1, \cdots, v_N\}$ represents the node set with $N$ nodes from disjoint $K$ classes, $\mathcal{E}^v$ is the edge set describing the connection relationships between pairwise nodes in the $v$-th view, and $\mathbf{X} \in \mathbb{R}^{N \times D}$ is the node attribute features. The connection relationships in the edge sets $\mathcal{E}^v$ can be mathematically formulated as the adjacency matrix $\mathbf{A}^v \in \{0, 1\}^{N \times N}$, where each element $\mathbf{A}_{ij}^v$ is defined as follows:

$$\mathbf{A}_{ij}^v = \begin{cases} 1, & \text{if } (v_i, v_j) \in \mathcal{E}^v, \\ 0, & \text{otherwise.} \end{cases} \tag{1}$$

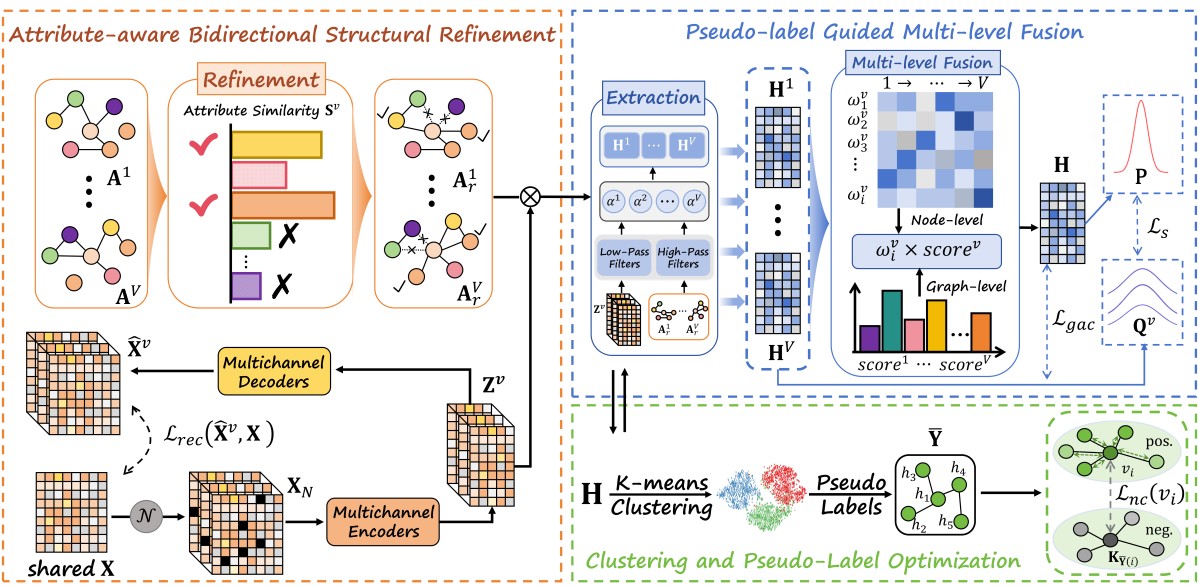

*Figure 1.* The overall framework of the proposed APGC method, which mainly consists of two core modules, namely ABSR module and PGMF. The ABSR module enhances attribute representations through the multi-channel autoencoders and performs dynamic bidirectional refinement of the original adjacency structure based on attribute similarity, thereby incorporating global attribute semantics into the structural modeling process. Furthermore, the PGMF module adopts a dual-channel filtering architecture to effectively capture intra-view consistent semantics as well as discriminative features. Under the guidance of clustering pseudo-labels, it further introduces a multi-level adaptive fusion mechanism to achieve fine-grained weighting and fusion at both the node-level and the view-level. Finally, through the joint optimization of multiple loss functions, the model is able to learn more discriminative consensus representation for custering.

The degree matrix is $\mathbf{D}^v = \mathbf{diag}(d_1^v, \cdots, d_N^v)$, where $d_i^v = \sum_{j=1}^{N} \mathbf{A}_{ij}^v$. The normalized graph Laplacian matrix is defined as $\widetilde{\mathbf{L}}^v = \mathbf{I} - \widetilde{\mathbf{A}}^v$, where $\widetilde{\mathbf{A}}^v = (\mathbf{D}^v)^{-1/2}\mathbf{A}^v(\mathbf{D}^v)^{-1/2}$. For clarity, the notations used in this study are summarized and explained in Appendix B.

## 2.2. Attribute-aware Bidirectional Structural Refinement

Existing DMGC methods are limited by the local static nature of the original adjacency matrix $\mathbf{A}^v$ where they struggle to capture global semantics and often overlook the role of node attribute representation. To address this, the proposed ABSR module introduces an attribute-aware bidirectional structural refinement strategy. This strategy dynamically achieves the topology augmentation based on attribute similarity, injecting global attribute semantics into the structural learning process while effectively suppressing structural noise. The details are introduced in the following section.

**Attribute Embedding Augmentation** Considering the high similarity typically exhibited by intra-class nodes in the attribute space, this paper introduces attribute similarity as a crucial basis for structural refinement. Before employing a multi-channel autoencoder to extract diverse and discriminative attribute embeddings, random noise perturbation is added on original attribute to improve robustness, thus com-

pelling encoder to learn more stable latent representations:

$$\mathbf{X}_N^v = \mathbf{X} + \mathcal{N}(0, \sigma_N^v) \tag{2}$$

where $\mathcal{N}(0, \sigma_N^v)$ denotes channel-specific Gaussian noise with a standard deviation of $\sigma_N^v$. Subsequently, attribute embeddings are extracted through the autoencoders:

$$\mathbf{Z}^v = f_\theta^v(\sigma(\mathbf{X}_N^v; \Theta^v)), \tag{3}$$

$$\widehat{\mathbf{X}}^v = g_\varphi^v(\sigma(\mathbf{Z}^v; \Phi^v)). \tag{4}$$

where $\sigma(\cdot)$ denotes the activation function. In addition, $f_\theta^v$ and $g_\varphi^v$ are the multi-channel encoders and decoders, while $\Theta^v$ and $\Phi^v$ represent the non-shared learnable parameters of the encoders and decoders, respectively. Here, augmented multi-view attribute embedding representations constructed from original single attribute aims to match multiple graphs, providing flexible and rich semantic alignment space.

To facilitate the fidelity of attribute representation learning in autoencoders $f_\theta^v(\cdot)$, a reconstruction loss is introduced between the original attribute features $\mathbf{X}$ and the reconstructed representations $\widehat{\mathbf{X}}^v$ to preserve feature consistency:

$$
\begin{aligned}
\mathcal{L}_{rec} &= \sum_{v=1}^{V} \mathcal{L}_{rec}^v(\widehat{\mathbf{X}}^v, \mathbf{X}) \\
&= -\sum_{v=1}^{V}(\mathbf{X} \log \widehat{\mathbf{X}}^v + (1 - \mathbf{X}) \log\left(1 - \widehat{\mathbf{X}}^v\right))
\end{aligned} \tag{5}
$$

**Bidirectional Structural Refinement** To fully exploit the potential global structural relationships embedded in the attribute features, the semantic correlation is computed based on the attribute embeddings as follows:

$$\mathbf{S}_{ij}^v = Sim\left(\mathbf{Z}_i^v, \mathbf{Z}_j^v\right), \forall i, j \in \{1, \cdots, N\} \quad (6)$$

where the function $Sim\left(\cdot, \cdot\right)$ measures the cosine distance in the vector space.

To overcome the limitations of locality inherent in the original static graph, this paper introduces an attribute similarity correlation matrix $\mathbf{S}$ to inject global semantic information. Based on this, the proposed bidirectional structural refinement strategy leverages attribute-induced semantic consistency to dynamically reconstruct the topology. By synergistically performing the addition of high-quality edges and the pruning of noisy ones, this mechanism effectively suppresses interfering connections while preserving reliable topological structures, ultimately achieving deep refinement and augmentation for original graph structure. It is formally defined as follows:

$$\mathbf{A}_{r,ij}^v = \begin{cases} 1, & \mathbf{S}_{ij}^v > \delta_u, \\ 0, & \mathbf{S}_{ij}^v < \delta_l, \\ A_{ij}^v, & others. \end{cases} \quad (7)$$

where $\mathbf{A}_{r,ij}^v$ denotes the refined node connection relationship between node $v_i$ and node $v_j$ in the $v$-view. To flexibly characterize edge quality, this paper introduces dual semantic thresholds, $\delta_u$ and $\delta_l$, to distinguish high-quality connections from noisy edges, respectively, while preserving the original structural information for connections with intermediate similarity. Specifically, when the attribute similarity of a node pair exceeds $\delta_u$, it is considered to possess strong semantic consistency, and the corresponding edge is reset to $1$. Conversely, when the similarity falls below $\delta_l$, it indicates significant semantic conflict, and the edge weight is suppressed to $0$. This strategy leverages attribute consistency constraints to compensate for the information deficiency of the static local topology. By dynamically supplementing potential semantic connections and pruning heterophilic pseudo-edges, it achieves bidirectional structural refinement and effectively enhances global relationship modeling capabilities.

### 2.3. Pseudo-label Guided Multi-level Fusion

**Single-view Information Extraction** Before performing multi-view fusion, the PGMF module first introduces a dual-channel filtering consisting of low-pass and high-pass filters to separately model intra-view consistent semantic information and discriminative features, thereby enabling comprehensive single-view information extraction. Specifically, a low-pass filter is constructed based on the graph Laplacian operator, which strengthens semantic consistency among

neighboring nodes through a neighborhood propagation mechanism, resulting in smooth and stable low-frequency representations. This process is defined as follows:

$$\begin{aligned} \mathbf{H}_L^{v,(l)} &= \left(\mathbf{I} - \widetilde{\mathbf{L}}_r^v\right) \mathbf{H}_L^{v,(l-1)} + \mathbf{Z}^v \\ &= \left(\mathbf{I} - \widetilde{\mathbf{L}}_r^v\right)\left(\left(\mathbf{I} - \widetilde{\mathbf{L}}_r^v\right) \mathbf{H}_L^{v,(l-2)} + \mathbf{Z}^v\right) + \mathbf{Z}^v \\ &\stackrel{\cdots}{=} \left(\mathbf{I} + \widetilde{\mathbf{A}}_r^v + \widetilde{\mathbf{A}}_r^{v^2} + \cdots + \widetilde{\mathbf{A}}_r^{v^l}\right) \mathbf{Z}^v \end{aligned}$$
$$(8)$$

where $l$ is the order of low-pass filtering, and $\mathbf{H}_L^{v,(l)}$ represents the feature representation after the $l$-th order low-pass filtering, with $v \in \{1, \cdots, V\}$. To enhance the stability of deep information propagation while preserving the original features, the model introduces residual connections at each filtering order. By integrating neighborhood information, the model is able to fully capture structural dependencies among nodes, thereby learning semantically consistent low-frequency feature representations.

In addition, to fully exploit the node-distinctive information, the PGMF module further introduces a high-pass filtering mechanism to extract high-frequency features. The high-pass filter is likewise constructed based on the graph Laplacian operator to emphasize inter-node differences, and its formulation is defined as follows:

$$\mathbf{H}_H^{v,(h)} = (\widetilde{\mathbf{L}}_r^v)\mathbf{H}_H^{v,(h-1)} = (\mathbf{I} - \widetilde{\mathbf{A}}_r^v)^h \mathbf{Z}^v \quad (9)$$

where $h$ is the order of high-pass filtering, and $\mathbf{H}_H^{v,(h)}$ represents the node feature representation after the $h$-th order high-pass filtering, with $v \in \{1, \cdots, V\}$. By progressively amplifying feature differences among nodes across layers, this process highlights structural variations, thereby effectively capturing high-frequency feature representations that reflect node heterogeneity and structural boundaries.

Then, a learnable balancing parameter $\alpha^v$ is introduced to adaptively adjust fusion weights between two components, yielding a comprehensive view-specific representation:

$$\mathbf{H}^v = \alpha^v \cdot \mathbf{H}_L^v + (1 - \alpha^v) \cdot \mathbf{H}_H^v \quad (10)$$

The $\mathbf{H}^v$ is capable of effectively modeling and integrating multi-frequency semantic information within the view.

**Multi-level Cross-view Fusion** Different views characterize the node from multiple perspectives, exhibiting both consistency and complementarity. To fully exploit the informational value of each view, PGMF designs a multi-level fusion strategy that performs weighted fusion of feature representations from different views at both the node-level and the view-level, exploiting cross-view consistency and complementarity at fine-grained level. This strategy captures the differentiated contributions of different views at same node while introducing view-level constraints at a global scale,

where local node-level adaptive weighting and global view-level weighting mechanisms collaboratively calibrate the specific and consistent contributions of each view for node-specific cross-view fusion. Finally, a more robust consensus representation $\mathbf{H}$ is computed as follows:

$$\mathbf{H}_{i,:} = \sum_{v=1}^{V} Norm \left( \underbrace{score^v}_{\text{view-level}} \cdot \underbrace{\omega_i^v}_{\text{node-level}} \right) \mathbf{H}_{i,:}^v \qquad (11)$$

where $\mathbf{H}_{i,:}$ is the final representation of node $v_i$, $\omega_i^v$ represents a learnable node-level fusion weight that characterizes the importance contribution of node $v_i$ in $v$-view to the corresponding consensus representation. The $score^v$ denotes the view-level weight obtained through an evaluation function, which globally adjusts the contribution proportion of different views to the final consensus representation in accordance with their assessed importance. The two components operate in a collaborative manner to realize fine-grained weighting and fusion at both the node-level and view-level, and the function $score^v$ is defined as follows:

$$score^v = eva(\mathbf{H}^v, \mathbf{H}), v = 1, 2, \cdots, V \qquad (12)$$

where $score^v$ quantifies the global importance of $v$-view using clustering accuracy. Specifically, Eq. (12) treats the clustering pseudo-labels generated from the consensus representation $\mathbf{H}$ as the ground truth to evaluate the clustering accuracy of the local representation $\mathbf{H}^v$. By synergizing node-level adaptive weights with view-level constraints, this strategy constructs a pseudo-label guided multi-level fusion mechanism, significantly enhancing the model's capability for dynamic integration of multi-view information during consensus representation learning.

To further enhance the collaborative fusion of multi-view representations, a weighted graph-level alignment induced contrastive loss $\mathcal{L}_{gac}$ is introduced. This loss explicitly incorporates view-level weights $score^v$. Guided by clustering pseudo-labels, these weights are used to adaptively regulate the constraint strength imposed by different views on the consensus representation during training. Hence, it achieves dynamic balancing of multi-view contributions and consistency optimization. It is defined as follows:

$$\mathcal{L}_{gac} = \frac{1}{V} \sum_{v=1}^{V} score^v \cdot MSE\left(\mathbf{H}^v, \mathbf{H}\right)$$

$$= \frac{1}{V(N \times d)} \sum_{v=1}^{V} score^v \sum_{i=1}^{N} \sum_{j=1}^{d} \left(\mathbf{H}_{ij}^v - \mathbf{H}_{ij}\right)^2 \qquad (13)$$

Subsequently, a node-level contrastive learning mechanism is introduced to further strengthen unsupervised representation learning in the latent space. Specifically, a positive

sample set is constructed based on the consensus representation $\mathbf{H}$ to enforce consistency among samples belonging to the same cluster in the latent space:

$$\Delta_i = \{j | j \in \text{top}(D(\mathbf{H}_{j,:}, \mathbf{H}_{i,:}), M), j \neq i\} \qquad (14)$$

Where $D(\cdot, \cdot)$ denotes the distance function, and $\Delta_i$ represents the set of the $\text{top } M$ most similar samples corresponding to sample $i$, which are regarded as the positive samples of $i$. For negative sample selection, we adopt a method inspired by that described in (Chao et al., 2024). Specifically, after a predefined number $T$ of training epochs, the cluster center representations $\{\mathbf{K}_1, \cdots, \mathbf{K}_K\}$ of different categories are used as negative samples. Based on this, the node-level contrastive learning loss is defined as follows:

$$l_{nc}(\mathbf{H}_{i,:}) = \log$$
$$\left( \frac{\sum\limits_{j \in \Delta_i} \exp\left(\frac{s(\mathbf{H}_{i,:}, \mathbf{H}_{j,:})}{\tau}\right)}{\sum\limits_{j \in \Delta_i} \exp\left(\frac{s(\mathbf{H}_{i,:}, \mathbf{H}_{j,:})}{\tau}\right) + \sum\limits_{k \neq \bar{\mathbf{Y}}(i)} \exp\left(\frac{s(\mathbf{H}_{i,:}, \mathbf{K}_k)}{\tau}\right)} \right) \qquad (15)$$

where clustering pseudo-labels $\bar{\mathbf{Y}}$ are obtained by applying K-means to the consensus representation $\mathbf{H}$. The overall node-level contrastive learning loss is formulated as follows:

$$\mathcal{L}_{nc} = -\frac{1}{N(K-1)} \sum_{k=1}^{K} \sum_{\{i|\bar{\mathbf{Y}}(i)=k\}} l_{nc}\left(\mathbf{H}_{i,:}\right) \qquad (16)$$

where $\bar{\mathbf{Y}}(i) = k$ indicates that node $v_i$ belongs to the $k$-cluster in pseudo-labels. Using non-corresponding clustering centers as negative samples can not only expand the perceptual domain of negative contrastive learning but also reduce computational complexity.

Meanwhile, although the self-supervised contrastive learning can effectively enhance the ability to discriminative representation learning, its optimization objective is not directly aligned with clustering performance. To address this limitation, the clustering-oriented self-supervised optimization loss is further introduced to guide the latent representations toward more favorable clustering structures:

$$\mathcal{L}_s = \sum_{v=1}^{V} score^v \text{KL}(\mathbf{P} \| \mathbf{Q}^v)$$
$$= \sum_{v=1}^{V} score^v \sum_i \sum_j p_{ij} \log \frac{p_{ij}}{q_{ij}^v} \qquad (17)$$

where $\text{KL}(\cdot \| \cdot)$ denotes the Kullback–Leibler divergence between two distributions, and $\mathbf{Q}^v$ is used to measure the similarity between the embedding representation $\mathbf{H}^v$ and the cluster centers:

$$q_{ij}^v = \frac{\left(1 + \|h_i^v - \mu_j^v\|^2\right)^{-1}}{\sum_{j'} \left(1 + \|h_i^v - \mu_{j'}^v\|^2\right)^{-1}} \qquad (18)$$

**Algorithm 1** Training Process of APGC.

**Input:** Multi-view graph $\mathcal{G}$; Cluster number $K$; Iteration number $It$; Filter orders $l$, $h$; Hyper-parameters $\alpha^v$, $\delta_u$, $\delta_l$, $M$, $T$.

**Output:** Clustering label matrix $\bar{\mathbf{Y}}$.

1: the Gaussian noise attribute $\mathbf{X}_N$ are obtained through Eq. (2).
2: **for** $iter = 1$ to $It$ **do**
3:   Multi-channel autoencoder is employed to obtain the attribute embeddings $\mathbf{Z}^v$ in Eq. (3).
4:   Calculate reconstruction loss $\mathcal{L}_{rec}$ by Eqs. (4)-(5).
5:   Obtain the similarity matrix $\mathbf{S}^v$ by calculating cosine similarity between attribute embeddings using Eq. (6).
6:   The $\mathbf{A}_r^v$ are obtained through the bidirectional structural refinement strategy in Eq. (7).
7:   Filters are designed to calculate the low-pass and high-pass representations $\mathbf{H}_L^v$ and $\mathbf{H}_H^v$ in Eqs. (8)-(9).
8:   Calculate the adaptive feature representation $\mathbf{H}^v$ for each view in Eq. (10).
9:   Obtain the consensus representation $\mathbf{H}$ using the multi-level fusion strategy in Eqs. (11)-(12).
10:   Calculate the graph-level alignment contrastive loss $\mathcal{L}_{gac}$ and the node-level contrastive loss $\mathcal{L}_{nc}$ in Eqs. (13)-(16).
11:   Calculate the self-supervised optimization loss $\mathcal{L}_s$ in Eqs. (17)-(19).
12:   Train the entire network by minimizing $\mathcal{L}$ in Eq. (20).
13: **end for**
14: Calculate the final clustering result $\bar{\mathbf{Y}}$ by performing K-means on $\mathbf{H}$.

where $q_{ij}^v$ is the probability that node $h_i^v$ belongs to cluster center $\mu_j^v$. The target distribution $\mathbf{P}$ is defined as follows:

$$p_{ij} = \frac{q_{ij}^2/f_j}{\sum_{j'} q_{ij'}^2/f_{j'}} \qquad (19)$$

where $f_j = \sum_i q_{ij}$ denotes the soft clustering frequency. By minimizing the KL divergence between $\mathbf{Q}^v$ and the target distribution $\mathbf{P}$, $\mathcal{L}_s$ encourages $\mathbf{Q}^v$ to converge toward $\mathbf{P}$, thereby promoting more compact clustering structures that better conform to target distribution characteristics.

### 2.4. Objective Function and Clustering

In summary, this paper proposes a novel DMGC framework APGC. Through the collaborative learning of the ABSR and PGMF modules, APGC achieves effective integration of multi-view information and embedding representation learning by jointly optimizing multiple loss functions. The overall training objective function of the proposed APGC

*Table 1.* The statistics of all used datasets.

| Dataset | $N$ | $D$ | $K$ | **Graphs** (Hom.ratio) |
|---------|-----|-----|-----|------------------------|
| ACM | 3025 | 1870 | 3 | $\mathcal{G}^1(0.82),\mathcal{G}^2(0.64)$ |
| DBLP | 4057 | 334 | 4 | $\mathcal{G}^1(0.80),\mathcal{G}^2(0.67),\mathcal{G}^3(0.32)$ |
| IMDB | 4780 | 1232 | 3 | $\mathcal{G}^1(0.48),\mathcal{G}^2(0.62),\mathcal{G}^3(0.40)$ |
| Texas | 183 | 1703 | 5 | $\mathcal{G}^1(0.09),\mathcal{G}^2(0.09)$ |
| Chameleon | 2277 | 2325 | 5 | $\mathcal{G}^1(0.23),\mathcal{G}^2(0.23)$ |
| Wisconsin | 251 | 1703 | 5 | $\mathcal{G}^1(0.19),\mathcal{G}^2(0.19)$ |

can be expressed as follows:

$$\mathcal{L} = \mathcal{L}_{rec} + \mathcal{L}_{gac} + \mathcal{L}_{nc} + \mathcal{L}_s \qquad (20)$$

where $\mathcal{L}$ denotes the overall training loss. After training, the K-means is directly applied to the final learned consensus representation $\mathbf{H}$ to obtain the clustering results $\bar{\mathbf{Y}}$. The detailed procedure of APGC is summarized in Algorithm 1.

## 3. Experiment

In this section, the superiority and effectiveness of the proposed APGC are validated through extensive experiments. The experimental environment is a desktop computer equipped with an Intel(R) Core(TM) i5-13490F CPU, a NVIDIA GeForce RTX 4060 GPU, 32GB RAM, and the PyTorch deep learning platform. Additional results are also shown in Appendix C.

### 3.1. Experimental Setting

#### 3.1.1. DATASETS

Following the key baselines in (Ling et al., 2023; Ren et al., 2025), the proposed APGC is evaluated on six homophilic and heterophilic graph datasets, including ACM (Fan et al., 2020), DBLP (Fan et al., 2020), IMDB (Fan et al., 2020), Texas (Rozemberczki et al., 2021), Chameleon (Rozemberczki et al., 2021) and Wisconsin (Pei et al., 2020). The detailed statistics of these datasets are briefly summarized in Table 1, where $N$, $D$, $K$ are the numbers of node, attribute feature dimension, class, respectively. And, Hom.ratio is the edge homophilic ratio of graph, where smaller means lower homophily.

#### 3.1.2. BASELINES

The proposed APGC is compared against 14 state-of-the-art baseline methods, including: Variational Graph Auto-Encoders (VGAE) (Kipf & Welling, 2016), Deep Attentional Embedding Graph Clustering (DAEGC) (Wang et al., 2019), Adaptive Graph Embedding (AGE) (Cui et al., 2020), O2MAC (Fan et al., 2020), MAGCN (Cheng et al., 2021), Attention-driven Graph Clustering Network (AGCN) (Peng et al., 2021), MCGC (Pan & Kang, 2021), Dual Correlation

*Table 2.* The performance comparison on six datasets. The Red and Blue values indicate the best and the suboptimal results, respectively.

| Dataset (Hom.ratio) | Metric | VGAE | DAEGC | AGE | O2MAC | MAGCN | AGCN | MCGC | DCRN | DuaLGR | BMGC | DIAGC | VGMGC | TRCGC | NGCE | OURS |
|---|---|---|---|---|---|---|---|---|---|---|---|---|---|---|---|---|
| ACM | ACC | 0.822 | 0.890 | 0.924 | 0.904 | 0.898 | 0.906 | 0.915 | 0.919 | 0.927 | 0.941 | 0.917 | 0.936 | 0.933 | 0.947 | 0.953 |
| (0.82 & 0.64) | NMI | 0.491 | 0.638 | 0.735 | 0.692 | 0.674 | 0.684 | 0.713 | 0.716 | 0.732 | 0.784 | 0.716 | 0.763 | 0.765 | 0.805 | 0.821 |
| | ARI | 0.544 | 0.701 | 0.789 | 0.739 | 0.721 | 0.742 | 0.763 | 0.776 | 0.794 | 0.833 | 0.770 | 0.819 | 0.813 | 0.850 | 0.864 |
| | F1 | 0.823 | 0.889 | 0.924 | 0.905 | 0.899 | 0.906 | 0.916 | 0.919 | 0.927 | 0.942 | 0.918 | 0.936 | 0.933 | 0.948 | 0.953 |
| DBLP | ACC | 0.886 | 0.665 | 0.753 | 0.907 | 0.928 | 0.733 | 0.930 | 0.797 | 0.924 | 0.940 | 0.933 | 0.932 | 0.931 | 0.933 | 0.942 |
| (0.80 & 0.67 & 0.32) | NMI | 0.693 | 0.308 | 0.450 | 0.729 | 0.772 | 0.397 | 0.830 | 0.490 | 0.755 | 0.801 | 0.783 | 0.783 | 0.786 | 0.791 | 0.806 |
| | ARI | 0.741 | 0.334 | 0.476 | 0.778 | 0.828 | 0.425 | 0.775 | 0.536 | 0.817 | 0.854 | 0.837 | 0.837 | 0.839 | 0.840 | 0.858 |
| | F1 | 0.874 | 0.656 | 0.746 | 0.901 | 0.923 | 0.728 | 0.925 | 0.793 | 0.918 | 0.936 | 0.928 | 0.927 | 0.931 | 0.928 | 0.938 |
| IMDB | ACC | 0.442 | 0.379 | 0.432 | 0.402 | 0.485 | 0.545 | 0.583 | 0.534 | 0.520 | 0.441 | 0.584 | 0.526 | 0.585 | 0.546 | 0.600 |
| (0.48 & 0.62 & 0.40) | NMI | 0.004 | 0.006 | 0.044 | 0.003 | 0.013 | 0.003 | 0.052 | 0.002 | 0.062 | 0.055 | 0.066 | 0.008 | 0.065 | 0.056 | 0.095 |
| | ARI | 0.009 | 0.010 | 0.046 | 0.002 | -0.018 | 0.014 | 0.103 | 0.001 | 0.125 | 0.049 | 0.132 | 0.032 | 0.136 | 0.127 | 0.191 |
| | F1 | 0.357 | 0.353 | 0.422 | 0.354 | 0.282 | 0.311 | 0.388 | 0.255 | 0.447 | 0.407 | 0.430 | 0.328 | 0.454 | 0.434 | 0.482 |
| Texas | ACC | 0.553 | 0.317 | 0.366 | 0.467 | 0.543 | 0.618 | 0.519 | 0.552 | 0.543 | 0.425 | - | 0.552 | - | 0.776 | 0.765 |
| (0.09 & 0.09) | NMI | 0.127 | 0.064 | 0.075 | 0.087 | 0.054 | 0.154 | 0.127 | 0.107 | 0.326 | 0.291 | - | 0.354 | - | 0.478 | 0.480 |
| | ARI | 0.217 | 0.026 | 0.073 | 0.146 | 0.011 | 0.181 | 0.129 | 0.151 | 0.260 | 0.158 | - | 0.260 | - | 0.549 | 0.597 |
| | F1 | 0.295 | 0.250 | 0.366 | 0.291 | 0.198 | 0.430 | 0.325 | 0.276 | 0.468 | 0.383 | - | 0.469 | - | 0.462 | 0.532 |
| Chameleon | ACC | 0.354 | 0.322 | 0.324 | 0.335 | 0.292 | 0.325 | 0.300 | 0.309 | 0.411 | 0.308 | - | 0.401 | - | 0.422 | 0.454 |
| (0.23 & 0.23) | NMI | 0.151 | 0.091 | 0.086 | 0.123 | 0.108 | 0.067 | 0.095 | 0.087 | 0.195 | 0.094 | - | 0.224 | - | 0.226 | 0.248 |
| | ARI | 0.124 | 0.056 | 0.076 | 0.089 | 0.033 | 0.061 | 0.059 | 0.057 | 0.160 | 0.059 | - | 0.134 | - | 0.193 | 0.180 |
| | F1 | 0.296 | 0.312 | 0.324 | 0.286 | 0.243 | 0.204 | 0.191 | 0.219 | 0.377 | 0.307 | - | 0.395 | - | 0.384 | 0.423 |
| Wisconsin | ACC | 0.493 | 0.327 | 0.311 | 0.400 | 0.477 | 0.498 | 0.518 | 0.502 | 0.564 | 0.515 | - | 0.566 | - | 0.737 | 0.801 |
| (0.19 & 0.19) | NMI | 0.105 | 0.106 | 0.093 | 0.110 | 0.081 | 0.064 | 0.129 | 0.108 | 0.341 | 0.340 | - | 0.416 | - | 0.468 | 0.544 |
| | ARI | 0.137 | 0.034 | 0.013 | 0.089 | 0.048 | 0.068 | 0.059 | 0.160 | 0.288 | 0.243 | - | 0.348 | - | 0.464 | 0.630 |
| | F1 | 0.341 | 0.283 | 0.311 | 0.279 | 0.206 | 0.249 | 0.307 | 0.341 | 0.471 | 0.408 | - | 0.496 | - | 0.472 | 0.676 |

Reduction Network (DCRN) (Liu et al., 2022), Dual Label-Guided Graph Refinement for Multi-View Graph Clustering (DuaLGR) (Ling et al., 2023), Balanced Multi-relational Graph Clustering (BMGC) (Shen et al., 2024), Dual Information Enhanced Multiview Attributed Graph Clustering (DIAGC) (Lin et al., 2025a), Variational Graph Generator for Multiview Graph Clustering (VGMGC) (Chen et al., 2025a), NGCE (Ren et al., 2025), Tnsor-Ring based Multi-view Contrastive Graph Clustering with High-quality Pseudo-labels (TRCGC) (Duan et al., 2026). Please refer to Appendix A or the original papers for a detailed description of these methods. For fair comparison, the original results of all competitors are directly taken from their respective papers.

### 3.2. Metrics

To comprehensively evaluate the clustering performance of all graph clustering methods, four widely used clustering metrics are utilized, including Accuracy (ACC), Normalized Mutual Information (NMI), Average Rand Index (ARI) and macro F1-score (F1). Each of these metrics is positively correlated with clustering performance.

### 3.3. Performance Comparison

The quantitative experimental results of the all comparative graph clustering methods on the benchmark datasets are shown in Table 2. It can be clearly observed that, except

for a very few metrics, APGC achieves optimal clustering performance on six homophilic and heterophilic datasets, demonstrating its superior generalization ability across different data environments. In homophilic scenarios such as ACM, DBLP, and IMDB datasets, APGC achieves average improvements of 5.8%, 2.7%, 5.9%, and 2.9% in ACC, NMI, ARI, and F1 metrics over representative BMGC (Shen et al., 2024). Specifically on the IMDB dataset, it outperforms the suboptimal method TRCGC (Duan et al., 2026) by 1.5%, 3.0%, 5.5%, and 2.8% on four metrics, respectively. More notably in challenging heterophilic scenarios such as Texas, Chameleon, and Wisconsin datasets, APGC achieves average gains of 2.8%, 3.3%, 6.7%, and 10.4% in the aforementioned four metrics compared to NGCE (Ren et al., 2025). Particularly on the structurally sparse Wisconsin dataset, APGC still achieves substantial improvements of 6.4%, 7.6%, 16.6%, and 20.4% respectively compared to NGCE (Ren et al., 2025). This superior performance is primarily attributed to the adaptive structure optimization and discriminative cross-view fusion mechanisms of APGC. Specifically, the ABSR module effectively breaks the limitations of static topology and homophily assumptions through dynamic bidirectional structural refinement, constructing view-oriented adjacency for powerful representation learning. And, the PGMF module generates discriminative consensus representation through cross-view and node-level collaborative multi-level adaptive fusion. Experimental results confirm the effectiveness of APGC in various DMGC

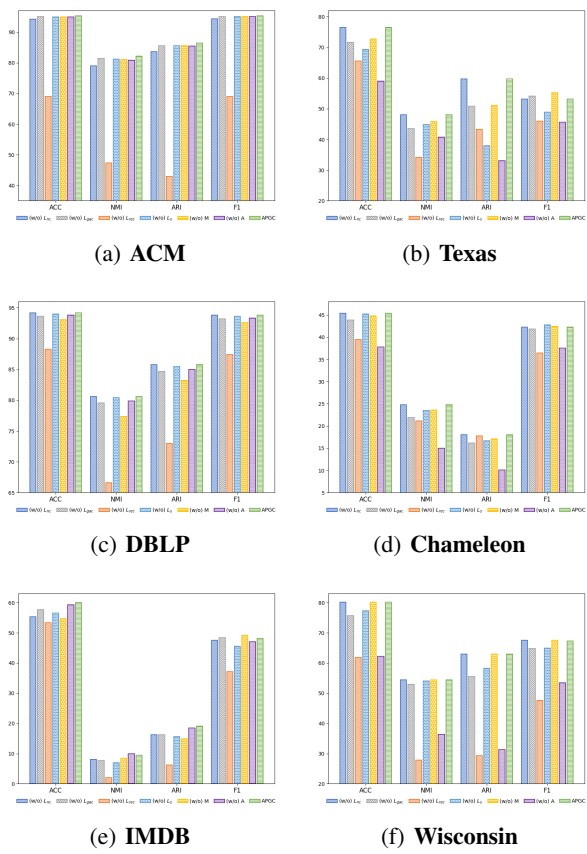

(a) **ACM**
(b) **Texas**
(c) **DBLP**
(d) **Chameleon**
(e) **IMDB**
(f) **Wisconsin**

*Figure 2.* The comparative clustering results of six ablation variants and APGC on the six used datasets.

task. More detailed analysis are provided in Appendix C.1.

### 3.4. Ablation Study

To systematically evaluate the contributions of each key component in APGC, six ablation variants are designed for comparison. Specifically, the variants are defined as follows. 1) **(w/o)** $\mathcal{L}_{nc}$: This variant removes the node-level contrastive loss $\mathcal{L}_{nc}$. 2) **(w/o)** $\mathcal{L}_{gac}$: This variant removes the graph-level alignment contrastive loss $\mathcal{L}_{gac}$. 3) **(w/o)** $\mathcal{L}_{rec}$: This variant removes the reconstruction loss $\mathcal{L}_{rec}$. 4) **(w/o)** $\mathcal{L}_s$: This variant removes the self-supervised loss $\mathcal{L}_s$. 5) **(w/o) A**: This variant does not employ the bidirectional structural refinement strategy to obtain $\mathbf{A}_r^v$. 6) **(w/o) M**: This variant fuses multi-view features through simple averaging strategy. From the experimental results shown in Figure 2, several notable observations can be made:

- On homophilic graphs, the performance of the **(w/o)** $\mathcal{L}_{rec}$ drops significantly, indicating that noisy attributes can introduce cumulative interference during information aggregation, whereas $\mathcal{L}_{rec}$ enhances attribute discriminability and provides more reliable inputs for aggregation. In addition, the **(w/o) M** also performs

poorly, suggesting that a single view-fusion strategy is insufficient to learn robust consensus representation.

- On heterophilic graphs, the performance of the **(w/o) A** degrades substantially, validating the critical role of bidirectional structural refinement in improving adjacency quality. Meanwhile, the **(w/o)** $\mathcal{L}_{rec}$ still exhibits inferior performance, indicating that unaugmented attribute features are insufficient to effectively support structural optimization.

- Overall, complete APGC consistently achieves the best performance, owing to the complementary synergy between ABSR and PGMF, where ABSR constructs high-quality multi-view graph for view-specific representation learning in PGMF, and PGMF optimizes the consensus embedding integration through multi-level fusion and contrastive learning.

### 3.5. Performance Comparison on Synthetic Datasets

Table 3 compares the results of five representative methods and APGC on six synthetic ACM datasets with different low homophilic ratios (Ling et al., 2023). As the homophilic ratio decreases from 0.50 to 0.00, the accuracy of traditional

*Table 3.* The performance comparison on the ACM dataset under different homophilic ratios. The best results are shown in **bold**.

| Datasets | Metric | VGAE | O2MAC | MAGCN | MCGC | DuaLGR | OURS |
|---|---|---|---|---|---|---|---|
| ACM (0.00 & 0.00) | ACC | 0.374 | 0.550 | 0.371 | 0.630 | 0.848 | **0.977** |
| | NMI | 0.005 | 0.250 | 0.009 | 0.498 | 0.551 | **0.901** |
| | ARI | 0.005 | 0.247 | 0.009 | 0.429 | 0.607 | **0.932** |
| | F1 | 0.371 | 0.546 | 0.355 | 0.535 | 0.845 | **0.977** |
| ACM (0.10 & 0.10) | ACC | 0.371 | 0.499 | 0.409 | 0.639 | 0.853 | **1.000** |
| | NMI | 0.005 | 0.176 | 0.019 | 0.529 | 0.559 | **1.000** |
| | ARI | 0.005 | 0.171 | 0.020 | 0.447 | 0.617 | **1.000** |
| | F1 | 0.356 | 0.497 | 0.391 | 0.546 | 0.850 | **1.000** |
| ACM (0.20 & 0.20) | ACC | 0.369 | 0.429 | 0.457 | 0.677 | 0.873 | **0.997** |
| | NMI | 0.004 | 0.096 | 0.053 | 0.291 | 0.592 | **0.979** |
| | ARI | 0.004 | 0.094 | 0.056 | 0.317 | 0.660 | **0.990** |
| | F1 | 0.349 | 0.428 | 0.454 | 0.672 | 0.871 | **0.997** |
| ACM (0.30 & 0.30) | ACC | 0.380 | 0.407 | 0.577 | 0.830 | 0.880 | **0.940** |
| | NMI | 0.007 | 0.067 | 0.154 | 0.518 | 0.602 | **0.770** |
| | ARI | 0.007 | 0.065 | 0.165 | 0.572 | 0.676 | **0.829** |
| | F1 | 0.376 | 0.405 | 0.577 | 0.829 | 0.880 | **0.940** |
| ACM (0.40 & 0.40) | ACC | 0.484 | 0.403 | 0.740 | 0.962 | 0.966 | **0.970** |
| | NMI | 0.097 | 0.055 | 0.369 | 0.839 | 0.851 | **0.869** |
| | ARI | 0.081 | 0.054 | 0.395 | 0.888 | 0.901 | **0.913** |
| | F1 | 0.490 | 0.402 | 0.742 | 0.962 | 0.966 | **0.970** |
| ACM (0.50 & 0.50) | ACC | 0.659 | 0.427 | 0.894 | 0.981 | 0.996 | **1.000** |
| | NMI | 0.262 | 0.066 | 0.646 | 0.910 | 0.978 | **1.000** |
| | ARI | 0.270 | 0.067 | 0.711 | 0.944 | 0.989 | **1.000** |
| | F1 | 0.664 | 0.426 | 0.894 | 0.981 | 0.996 | **1.000** |

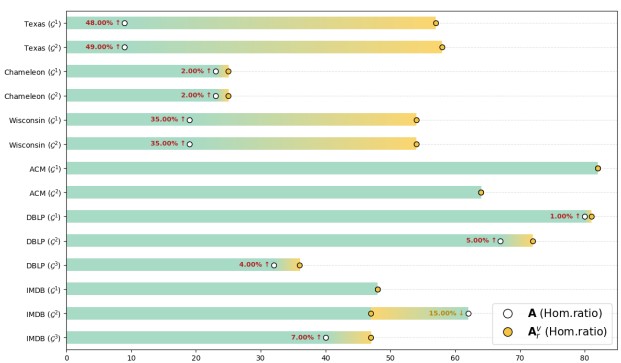

*Figure 3.* The comparison of the homophily ratio for the $\mathbf{A}$ and the $\mathbf{A}_r^v$ on six datasets.

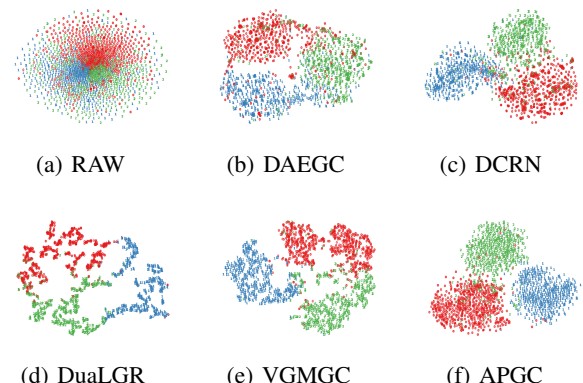

*Figure 4.* The 2-D visualization on ACM dataset. The 2-D visualization of the DBLP dataset is provided in Appendix C.5.

methods such as MAGCN (Cheng et al., 2021) plummets by over $50\%$ due to structural noise interference, whereas APGC maintains a high accuracy of $97.7\%$. This robust performance is attributed to the unique dynamic bidirectional refinement mechanism of the ABSR module. By leveraging augmented attribute representations to guide topological reconstruction, ABSR effectively eliminates semantic conflicts arising from heterophilic edges and compensates for the deficiency of structural information. This strategy of synergistically optimizing global attribute semantics with local topology enables APGC to achieve effective multi-view fusion and precise clustering. Even in scenarios with extremely low homophily (Hom.ratio = 0), the APGC still achieves satisfactory performance, verifying its significant scalability in complex heterophilic graph.

### 3.6. Analysis of Structural Homophily

To further validate the effectiveness of the ABSR module in structural augmentation, the homophily ratios of the original adjacency matrix $\mathbf{A}$ and the refined structural matrix $\mathbf{A}_r^v$ are visually compared. As shown in Figure 3, the homophily of most views achieved significant improvement after the bidirectional structural refinement strategy. Specifically, on the DBLP dataset with inherently high homophily, the ABSR provides further augmentation, boosting the homophily ratios of $\mathcal{G}^1$, $\mathcal{G}^2$, and $\mathcal{G}^3$ by $1\%$, $5\%$, and $4\%$ respectively compared to the original structures. Conversely, on the low-homophily Texas dataset, this strategy demonstrate strong structural reconstruction capabilities, where the homophily ratios for $\mathcal{G}^1$ and $\mathcal{G}^2$ achieve massive increases of $48\%$ and $49\%$, respectively. These results strongly indicate that while augmenting attribute embeddings, the ABSR module can leverage the attribute semantics to precisely introduce high-quality connections and suppress heterophilic edges. In summary, the ABSR module demonstrates the clear superiority in structural augmentation, effectively improving the topological quality, especially for heterophilic graph.

### 3.7. Visualization Analysis

To intuitively demonstrate the advantages of APGC in discriminative representation learning, the raw attribute feature and embedding representations learned by several representative DMGC methods are visualized using t-SNE (Maaten & Hinton, 2008). As shown in Figure 4 and Appendix C.5, different clusters are marked with distinct numbers and colors. It can be observed that the distribution of nodes with raw features appears irregular. Among all DMGC methods, APGC exhibits the strongest discriminability in embedding representation, as evidenced by nodes within the same cluster being more compact and boundaries between different clusters being relatively well-defined.

## 4. Conclusions

To address the issues of over-reliance on static topology in heterophilic scenarios and coarse-grained fusion strategies prevalent in existing DMGC methods, this paper proposed a novel framework named APGC. By introducing ABSR module, APGC leveraged diverse global attribute semantics to dynamically reconstruct the graph structure. It precisely eliminated semantic conflict edges while reinforcing high-quality connections, thereby effectively overcoming the limitations of traditional homophily assumptions. Furthermore, PGMF module innovatively achieved adaptive synergy fusion at both node and view levels. Coupled with a dual contrastive learning mechanism, it significantly improved the discriminability of the consensus representation. Extensive experimental results on six homophilic and heterophilic datasets demonstrated that APGC significantly outperforms current state-of-the-art methods in clustering performance. Of course, exploring more refined high- and low-frequency fusion mechanisms and how to tackle extremely sparse graph data are two research directions worth investigating.

## Acknowledgment

This research was supported by the National Natural Science Foundation of China under Grant 62403043, 62225303, and 62433004; in part by the Beijing Natural Science Foundation under Grant L2603012; in part by the Interdisciplinary Research Center of Beijing University of Chemical Technology under Grant XK2025-06.

## Impact Statement

This paper presents work whose goal is to advance the field of Machine Learning. There are many potential societal consequences of our work, none which we feel must be specifically highlighted here.

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

## A. Related Studies

With the rapid advancement of deep learning technologies, GNN-based DMGC methods have evolved from the exploration of foundational architectures to the deepening of complex mechanisms. Early pioneers like One2Multi graph Autoencoder Clustering (O2MAC) (Fan et al., 2020) innovatively utilized graph autoencoders to reconstruct multi-view attributes and structures to capture shared latent representations. Multi-view Attribute Graph Convolution Networks for clustering (MAGCN) (Cheng et al., 2021) established the foundation for the joint modeling of cross-view information by designing a dual-path encoding architecture that separately models intra-view features and inter-view consistency. Subsequent research has been dedicated to optimizing the efficiency of inter-view information interaction where Dual Information enhanced multiview Attributed Graph Clustering (DIAGC) (Lin et al., 2025a) introduced a decoupling mechanism to separate consensus information from view-specific information, thereby enabling the model to focus on more discriminative features. Addressing the issue of uneven multi-view contributions, Balanced Multi-relational Graph Clustering (BMGC) (Shen et al., 2024) balanced the weights of different views by mining dominant views within multi-relational graphs to effectively mitigate the negative impact of informational discrepancies on clustering performance. To further enhance the discriminability of embedding representations, self-supervised contrastive learning mechanisms have been widely adopted. For instance, contrastive Multi-View Representation Learning on Graphs (MVGRL) (Hassani & Khasahmadi, 2020) and Multi-view Contrastive Graph Clustering (MCGC) (Pan & Kang, 2021) leart consistent node representations by maximizing mutual information across views or scales, yielding significant improvements in model performance. Meanwhile, multi-view graph clustering via Node-Guided Contrastive Encoding (NGCE) (Ren et al., 2025) approached the problem from a node-guidance perspective by attempting to utilize node features to guide embedding generation for integrating heterophilic information, aiming to overcome the limitations of traditional methods on non-homophilic graphs. Furthermore, the introduction of pseudo-label guidance has become a crucial strategy for boosting clustering performance. Specifically, Dual Label-Guided graph Refinement for multi-view graph clustering (DuaLGR) (Ling et al., 2023) iteratively refined graph structures through a dual pseudo-label mechanism. Tensor-Ring based multi-view Contrastive Graph Clustering with high-quality pseudo-labels (TRCGC) (Duan et al., 2026) introduced tensor ring decomposition combined with a reward mechanism to select high-quality pseudo-labels, effectively improving the convergence performance of unsupervised model training.

Despite these advancements, existing methods still face two major bottlenecks regarding their reliance on static topological structures and rigid fusion strategies. In contrast, the proposed APGC effectively resolves heterophilic interference and uneven fusion issues to achieve the discriminative representation learning through the dynamic bidirectional structural refinement of the ABSR module and the adaptive fusion across node-view levels of the PGMF module.

## B. Notations

This section summarizes and explains the notations used in this study in Table 4.

## C. Additional Experiments

In this section, additional experiments are conducted to further verify the effectiveness of the proposed APGC. These include a detailed analysis of the performance comparison experiment (Section C.1, Section C.2), computational complexity (Section C.3), analysis of parameters $\delta_u$ and $\delta_l$ (Section C.4), as well as comprehensive visualization results (Section C.5).

### C.1. Performance Comparison

The quantitative experimental results of the all graph clustering comparative methods on the benchmark datasets are shown in Table 2. In the table, red values denote the best clustering results, while blue values denote the sub-optimal results.

Except for a very small number of metrics, APGC achieves state-of-the-art clustering performance on six homophilic and heterophilic datasets, fully demonstrating its strong generalization ability across diverse data environments. Taking the IMDB dataset as an example, APGC outperforms the suboptimal TRCGC (Duan et al., 2026) by 1.5%, 3.0%, 5.5%, and 2.8% in terms of ACC, NMI, ARI, and F1, respectively. This indicates that APGC can more effectively capture latent semantic consistency across multiple views and learn the more discriminative consensus representation. On the Wisconsin dataset, which is characterized by sparse structures and significant noise, APGC still exhibits stable and pronounced performance advantages, surpassing NGCE (Ren et al., 2025) by 6.4%, 7.6%, 16.6%, and 20.4% on ACC, NMI, ARI, and F1, respectively. These results systematically verify that APGC consistently maintains superior clustering performance

*Table 4.* The explanation of main notations.

| Notation | Meaning |
|---|---|
| $\mathbf{X} \in \mathbb{R}^{N \times D}$ | Original attribute feature |
| $\widehat{\mathbf{X}}^v \in \mathbb{R}^{N \times D}$ | Reconstructed attribute feature |
| $\mathbf{Z}^v \in \mathbb{R}^{N \times d}$ | Attribute embedding representation |
| $\mathbf{A}^v \in \{0,1\}^{N \times N}$ | Original adjacency matrix |
| $\mathbf{A}_r^v \in \{0,1\}^{N \times N}$ | refined structural matrix |
| $\mathbf{S}^v \in \{0,1\}^{N \times N}$ | Semantic similarity matrix |
| $\mathbf{D}^v \in \mathbb{R}^{N \times N}$ | Degree matrix |
| $\widetilde{\mathbf{L}}^v \in \mathbb{R}^{N \times N}$ | Graph Laplacian matrix |
| $\mathbf{H}_L^{v,(l)} \in \mathbb{R}^{N \times d}$ | Low-pass filtering representation |
| $\mathbf{H}_H^{v,(h)} \in \mathbb{R}^{N \times d}$ | High-pass filtering representation |
| $\mathbf{H}^v \in \mathbb{R}^{N \times d}$ | Feature representation of $v$-th view |
| $\mathbf{H} \in \mathbb{R}^{N \times d}$ | Consensus feature representation |
| $\mathbf{I} \in \{0,1\}^{N \times N}$ | Identity matrix |
| $\Delta_i \subseteq \mathcal{V}, |\Delta_i| = M$ | Positive sample set of node $v_i$ |

across different data scenarios.

Further, several fine-grained analyses are drawn from the following aspects:

1) On homophilic graph datasets such as ACM, DBLP, and IMDB, APGC achieves clustering performance that is significantly superior to existing methods across all evaluation metrics. Taking the representative method BMGC (Shen et al., 2024) as an example, APGC attains average improvements of $5.8\%$, $2.7\%$, $5.9\%$, and $2.9\%$ in terms of ACC, NMI, ARI, and F1, respectively, demonstrating its effectiveness in homophilic structural scenarios. These performance gains mainly stem from attribute augmentation and multi-view fusion. On the one hand, ABSR incorporates node attribute information and leverages reconstruction learning under noise perturbations to extract discriminative attribute embeddings while injecting global attribute semantics. On the other hand, PGMF performs adaptive weighted fusion at both the node-level and the view-level, and further enhances multi-level semantic alignment through contrastive learning, resulting in more discriminative consensus representations.

2) On heterophilic graph datasets such as Texas, Chameleon, and Wisconsin, APGC likewise exhibits clustering performance that is significantly superior to existing methods. In contrast, traditional models that rely on the homophily assumption generally suffer from pronounced performance degradation. Compared with the representative method NGCE (Ren et al., 2025), APGC achieves average improvements of $2.8\%$, $3.3\%$, $6.7\%$, and $10.4\%$ in terms of ACC, NMI, ARI, and F1, respectively. This result indicates that APGC can maintain stable and superior clustering performance even in complex scenarios where the homophily assumption does not hold. These advantages mainly stem from the attribute-aware bidirectional structural refinement strategy introduced in APGC. On the one hand, ABSR selectively strengthens high-quality connections with strong semantic consistency, compensating for information loss caused by structural sparsity or erroneous links. On the other hand, it effectively suppresses semantically conflicting edges induced by heterophilic relations, thereby dynamically injecting global attribute semantics into local topological structures. This mechanism overcomes the reliance of traditional GNNs on static structures and the homophily assumption, enabling the model to learn high-quality and highly discriminative node representations even on heterophilic graphs with unreliable structures.

In conclusion, these experimental results and analyses evidently validate the effectiveness of the proposed APGC method in complex DMGC task.

### C.2. Performance comparison on the Amazon dataset

Benefiting from low-dimensional computations and parallelizable design of K-means, the model exhibits strong potential for large-scale scalability, which is further validated by the newly added experiments on Amazon dataset in Table 5. The proposed APGC can still achieve excellent performance on complex Amazon.

*Table 5.* Performance comparison on the Amazon dataset. Best results are in bold.

| Dataset | Metric | VGAE | O2MAC | MAGCN | MCGC | DuaLGR | BMGC | OURS |
|---------|--------|-------|-------|-------|-------|--------|-------|-------|
| Amazon | ACC | 0.319 | 0.443 | 0.519 | 0.468 | 0.612 | 0.786 | **0.802** |
| | NMI | 0.016 | 0.134 | 0.232 | 0.215 | 0.277 | **0.577** | 0.576 |
| | ARI | 0.013 | 0.090 | 0.114 | 0.106 | 0.272 | 0.563 | **0.574** |
| | F1 | 0.273 | 0.442 | 0.507 | 0.480 | 0.622 | 0.785 | **0.803** |

## C.3. Computational complexity

Specifically, the complexity of the autoencoder in the ABSR module is $\mathcal{O}(NDd)$. Although the similarity computation has a theoretical complexity of $\mathcal{O}(N^2 d)$, performing it in a low-dimensional latent space reduces the actual computational burden. The bidirectional structural refinement has a complexity of $\mathcal{O}(N^2)$. For the PGMF module, the complexities of the dual-channel filtering and multi-level weighting are approximately $\mathcal{O}(N^2 d)$ and $\mathcal{O}(Nd)$, respectively. Combined with the loss function of $\mathcal{O}(N(D + 2d))$, the overall complexity of APGC is approximately $\mathcal{O}(N^2 d)$. This is comparable to state-of-the-art baselines (e.g., NGCE), yet APGC achieves superior clustering performance. We provide a comparison of the running time per epoch across two classic datasets in Table 6.

*Table 6.* Comparison of the running time per epoch on ACM and DBLP datasets.

| Dataset | Metric | DCRN | DuaLGR | DIAGC | VGMGC | NGCE | OURS |
|---------|--------|------|--------|-------|-------|------|------|
| ACM | ms | 50 | 10 | 90 | 780 | 480 | 80 |
| DBLP | ms | 80 | 10 | 210 | 1230 | 860 | 160 |

## C.4. Parameter Analysis

This section conducts an in-depth analysis of the key hyperparameters in APGC, focusing on how the model's performance varies with the parameter $\delta_u$ and $\delta_l$ in the ABSR. Additionally, the distribution of learnable parameter $\alpha^v$ is also discussed.

### C.4.1. SENSITIVITY ANALYSIS OF $\delta_u$

The parameter $\delta_u$ serves as the critical similarity threshold for identifying high-quality connections within the bidirectional structural refinement mechanism, with its candidate range set to $\{0.1, 0.2, 0.3, 0.4, 0.5, 0.6, 0.7, 0.8, 0.9, 0.9999\}$. As shown in Figure 5, model performance exhibits significantly distinct trends as $\delta_u$ varies. On homophilic datasets such as ACM and DBLP, clustering performance demonstrates a trend of initially rising and then stabilizing as $\delta_u$ increases, achieving optimality in the high-threshold range of $\{0.8, 0.9, 0.9999\}$. This phenomenon is primarily attributed to the inherent high topological homophily of these datasets, where lower thresholds tend to introduce noisy pseudo-edges. Only under strict threshold constraints, the model can precisely capture high-confidence semantic connections while filtering out noise, thereby effectively enhancing the overall topological quality. Conversely, on heterophilic datasets, model performance exhibits a rise-then-fall trend and maintains a high level within the intermediate range of $\{0.6, 0.7, 0.8, 0.9\}$. This result indicates that heterophilic graph structures are more sensitive to the threshold. A moderate threshold can introduce sufficient intra-cluster semantic connections to alleviate structural heterophily while suppressing noise. Conversely, an excessively high threshold leads to an insufficient number of effective supplemented edges, thereby limiting the ability of structural refinement to recover latent homophilic relationships.

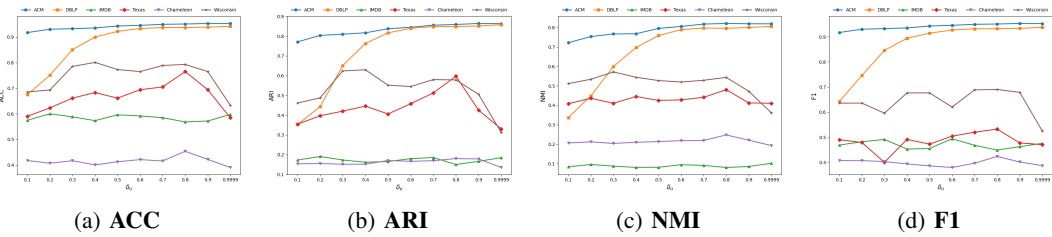

| (a) **ACC** | (b) **ARI** | (c) **NMI** | (d) **F1** |

*Figure 5.* The clustering results of APGC with different $\delta_u$ on six datasets.

### C.4.2. SENSITIVITY ANALYSIS OF $\delta_l$

The parameter $\delta_l$ serves as the critical threshold for suppressing noisy edges within the bidirectional structural refinement mechanism, with its candidate range set to $\{-0.9, -0.8, -0.7, -0.6, -0.5, -0.4, -0.3, -0.2, -0.1, 0.0\}$. As shown in Figure 6, the clustering performance exhibits a slight decline when $\delta_l$ is at $\{-0.1, 0.0\}$. This is primarily attributed to the fact that similarity intervals approaching 0 often correspond to semantically ambiguous decision boundaries (i.e., hard samples), where the forcible pruning of such connections may lead to the loss of potentially valid structural information. This result strongly validates the necessity and rationality of the proposed strategy in preserving original connections within the intermediate similarity interval. Overall, with the exception of minor fluctuations at the aforementioned boundaries, the model maintains high stability across a wide parameter range, demonstrating excellent robustness and insensitivity to $\delta_l$ across all clustering metrics on different datasets.

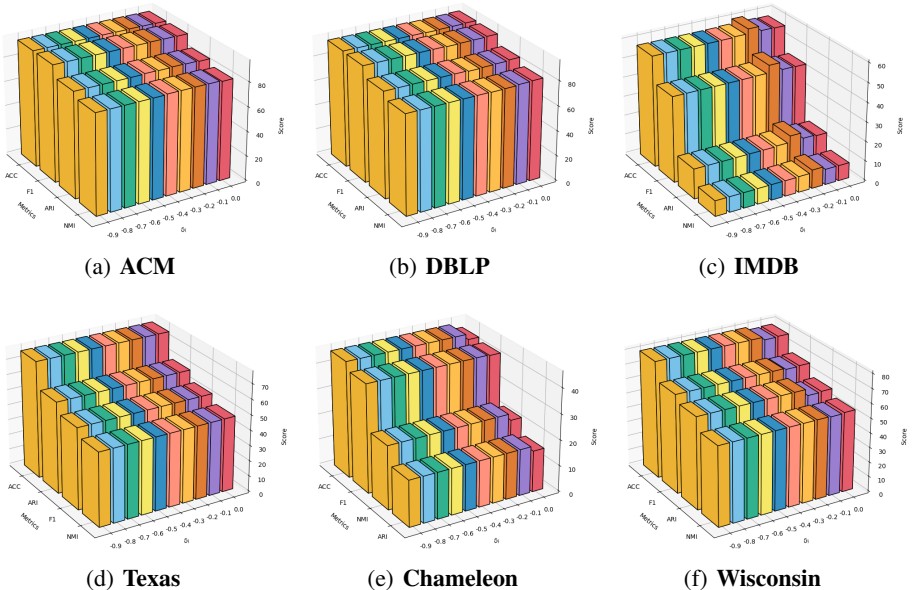

| (a) **ACM** | (b) **DBLP** | (c) **IMDB** |
|---|---|---|
| (d) **Texas** | (e) **Chameleon** | (f) **Wisconsin** |

*Figure 6.* The clustering results of APGC with different $\delta_l$ on six datasets.

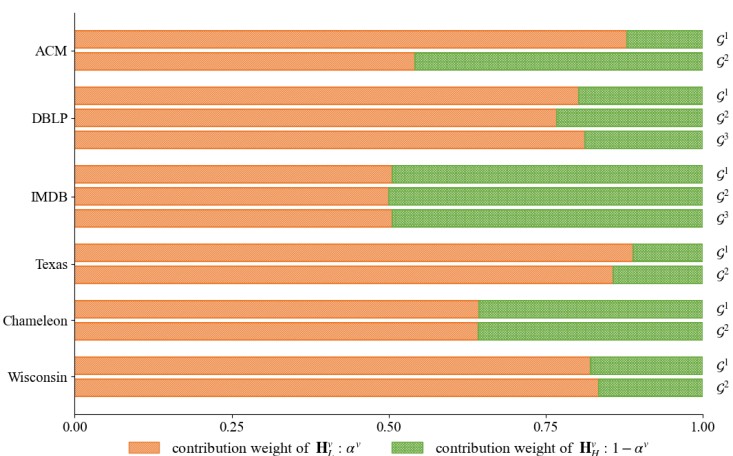

*Figure 7.* The visualization of the learnable balancing parameter $\alpha^v$ on six datasets.

### C.4.3. ANALYSIS OF LEARNABLE PARAMETER $\alpha^v$

To intuitively demonstrate the crucial role of high-frequency information in graph representation learning, this paper conducts a visual analysis of the learnable balance weight $\alpha^v$ as shown in Figure 7. The results confirm that high-frequency information

is equally indispensable for generating discriminative representation. While low-frequency features effectively preserve the global overall structure and common information, high-frequency features possess unique advantages in capturing local node discrepancies and delineating inter-cluster boundaries, thereby significantly enhancing inter-cluster separability. APGC employs a deep integration mechanism of multi-frequency features by adaptively weighting both components via the learnable parameter $\alpha^v$. By fully leveraging the complementary value of high-frequency information, this strategy substantially improves the discriminability and comprehensive expressive capability of the consensus representation. In addition, in most cases, low-frequency information dominates the main contribution, even in heterophilic graphs. This is mainly due to the fact that the ABSR module can correct the graph structure and improve its homophily connections, so the neighboring information aggregation is still important.

## C.5. Visualization

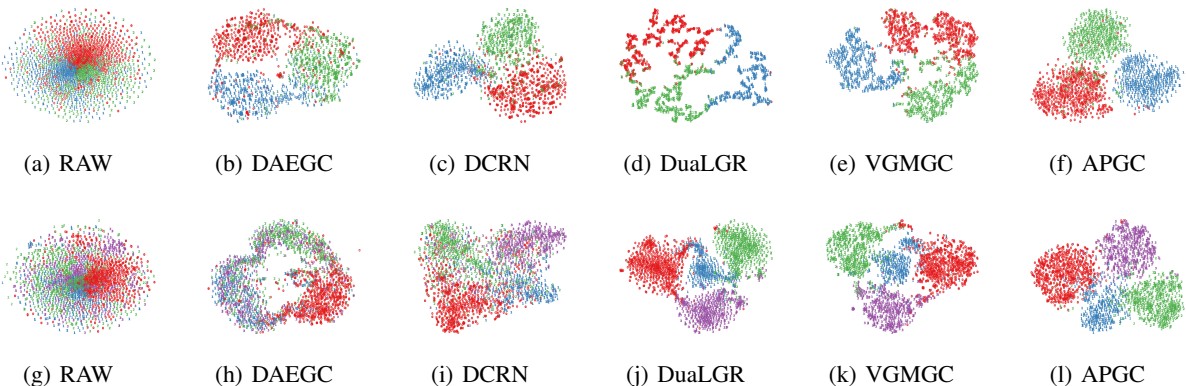

| (a) RAW | (b) DAEGC | (c) DCRN | (d) DuaLGR | (e) VGMGC | (f) APGC |
| (g) RAW | (h) DAEGC | (i) DCRN | (j) DuaLGR | (k) VGMGC | (l) APGC |

*Figure 8.* The 2-D visualization on two datasets. The first row and second row correspond to ACM and DBLP datasets, respectively.

