# OpenReview forum: "Deep Multi-view Graph Clustering via Attribute-aware Bidirectional Structural Refinement and Pseudo-label Guided Multi-level Fusion"
_ICML.cc/2026/Conference — ICML 2026 regular_

### Official Review · Reviewer_ozZ5 · 2026-03-11

**Soundness:** 3
**Presentation:** 3
**Significance:** 3
**Originality:** 3
**Overall Recommendation:** 5
**Confidence:** 5

**Summary:**

This paper presents a new deep multi-view graph clustering framework called APGC, which is designed to overcome the limitations of existing methods that depend heavily on static graph structures and use simplistic fusion strategies. The experiment results show that APGC delivers excellent clustering performance on a range of homophilic and heterophilic graph datasets.

**Compliance With Llm Reviewing Policy:**

Affirmed.

**Final Justification:**

The authors' responses have addressed my concerns, and I tend to accept this paper. So I would like to raise the score from 4 to 5.

**Key Questions For Authors:**

1. If the original graph $\mathbf{A}^v$ is inherently of low quality, the intermediate interval defined by $\delta_l$ and $\delta_u$ may still contain considerable topological noise. In that case, would directly preserving the corresponding connections in $\mathbf{A}^v$ introduce this noise into the model and thus impact the final learned representations?

2. Why are the values simply set to 1 and 0 during structure refinement, rather than using more precise values such as the actual attribute similarity?

3. During neighbor aggregation, the study applies a filtering approach based on the Laplacian operator. What are the main advantages of this method compared to other graph neural networks like GCN?

4. In fusing high- and low-frequency information, the study uses $\alpha^v$ for weighted fusion. In contrast, graph attention mechanisms (e.g., GAT) can adaptively learn importance weights for different nodes or neighbors. What motivated the choice of $\alpha^v$ instead of an attention mechanism?

5. In Experiment 3.4, only results under certain homophily levels are reported, with missing data for homophily ratios 0.3 and 0.4. Providing experimental results for this range could help more comprehensively analyze how the model performs under varying degrees of homophily.

**Limitations:**

The paper requires the discussion on limitations, such as structural noise processing and high-low frequency fusion.

**Strengths And Weaknesses:**

Strengths:

1. This paper effectively strengthens homophilic links and reduces heterophilic noise by dynamically rebuilding the graph structure based on attribute similarity. It also implements adaptive fusion spanning from nodes to views.
2. The work shows considerable innovation and makes solid technical contributions.
3. The experiments are well-designed and thorough, producing convincing outcomes.

Weaknesses:

The paper has some limitations. Areas such as handling structural noise and integrating high- and low-frequency information could be further improved.

---

> ### Author Rebuttal · Authors · 2026-03-29
>
> **Thank you for your thoughtful and detailed review. We will carefully address your issue below.**
>
> **Q1: Structural noise.**
>
> **A1**: Even with suboptimal original graphs, retaining connections between $\delta_u$ and $\delta_l$ introduces minor topological noise but does not severely harm representation learning. First, retaining these semantically ambiguous hard samples prevents losing valid connections, balancing denoising with information preservation. Second, the bidirectional structural refinement uses attributes to enhance topology, directly reconstructing high-confidence connections and significantly boosting the homophily ratio (see Section 3.5). Finally, the highly robust PGMF module leverages node-level and view-level adaptive weights for multi-level fusion, dynamically evaluating and effectively diluting local noise interference through cross-view collaboration.
>
> **Q2: Value assignment in structural refinement.**
>
> **A2**: Adopting a hard 0 and 1 strategy aims to utilize attributes to enhance high-quality connections and suppress semantic conflicts, achieving a structural-attribute complement. Furthermore, directly assigning continuous attribute similarities would easily degenerate the original topology into a pure attribute similarity graph, completely discarding its unique structural semantics.
>
> **Q3: Neighboring aggregation strategy.**
>
> **A3**: Classic GCN are essentially merely single low-pass filters. In contrast, our dual-channel filtering mechanism can explicitly decouple and simultaneously extract high- and low-frequency information, thereby capturing richer node features. Furthermore, adopting a parameter-free Laplacian filter strips away the cumbersome feature transformation matrices, which improves the computational efficiency of neighborhood information aggregation to a certain extent.
>
> **Q4: High- and low-frequency fusion.**
>
> **A4**: First, utilizing $\alpha^v$ directly and effectively controls the frequency preference of the view, thereby extracting its global information. Introducing GAT at this stage would inevitably incur additional feature transformation matrices and complex attention computations. Furthermore, in the subsequent multi-level fusion stage, the PGMF module has already incorporated fine-grained and node-level adaptive weights. Thus, leveraging $\alpha^v$ for view-level frequency balancing, complemented by node-level weights for feature fusion, synergistically builds a highly efficient and sufficiently adaptive fusion architecture. Admittedly, compared to the straightforward use of $\alpha^v$, exploring more sophisticated fusion mechanisms remains an direction for future. We will add a discussion on this aspect in our revised manuscript.
>
> **Q5: Additional experiments.**
>
> **A5**: We sincerely thank the reviewer for this question. As shown in the table, we have supplemented the experimental results with homophily ratios of 0.3 and 0.4. The results demonstrate that APGC still exhibits satisfactory performance, proving its generalizability across different heterophilic graphs. Especially in low homophily ratios, the performance advantage of the proposed APGC method is more obvious, fully verifying its effectiveness and stong universality in both homophilic and heterophilic graphs.
>
> | Datasets | Metric | VGAE | O2MAC | MAGCN | MCGC | DuaLGR | OURS |
> | :--- | :--- | :---: | :---: | :---: | :---: | :---: | :---: |
> | ACM (0.30 & 0.30) | ACC | 0.380 | 0.407 | 0.577 | 0.830 | 0.880 | **0.940** |
> | | NMI | 0.007 | 0.067 | 0.154 | 0.518 | 0.602 | **0.770** |
> | | ARI | 0.007 | 0.065 | 0.165 | 0.572 | 0.676 | **0.829** |
> | | F1 | 0.376 | 0.405 | 0.577 | 0.829 | 0.880 | **0.940** |
> | ACM(0.40 & 0.40)| ACC | 0.484 | 0.403 | 0.740 | 0.962 | 0.966 |**0.970** |
> | | NMI | 0.097 | 0.055 | 0.369 | 0.839 | 0.851 | **0.869** |
> | | ARI | 0.081 | 0.054 | 0.395 | 0.888 | 0.901 | **0.913** |
> | | F1 | 0.490 | 0.402 | 0.742 | 0.962 | 0.966 | **0.970** |
>
> **W1: Some limitations.**
>
> **A6**: Since the limitations regarding the handling of structural noise and the integration of high- and low-frequency information have already been comprehensively answered in Q1 and Q4, we refrain from reiterating them here.

---

> > ### Author Rebuttal · Reviewer_ozZ5 · 2026-04-03
> >
> > Thanks for the feedback. They have addressed my concerns.

---

> > > ### Author Response · Authors · 2026-04-03
> > >
> > > Thank you for your positive feedback. We sincerely thank you once again for your meticulous review and the valuable suggestions to improve our work.

---

### Official Review · Reviewer_NgqX · 2026-03-12

**Soundness:** 3
**Presentation:** 3
**Significance:** 3
**Originality:** 4
**Overall Recommendation:** 5
**Confidence:** 4

**Summary:**

This paper proposes a deep graph clustering method that enhances clustering performance by introducing attribute-aware bidirectional structural refinement and pseudo-label guided multi-level fusion. By leveraging node attribute similarity to optimize the original graph structure, it reduces structural noise and strengthens semantic relationships. Subsequently, during the representation learning phase, both low-pass and high-pass information are integrated to extract node features. Pseudo labels are then employed to adaptively fuse information from different views at both the node and view levels, yielding more representative consensus representations. Experiments across multiple datasets demonstrate superior performance compared to existing methods. The paper exhibits a well-structured framework and rich experimental design.

**Compliance With Llm Reviewing Policy:**

Affirmed.

**Final Justification:**

Given the paper and the rebuttal, I would like to keep my current score.

**Key Questions For Authors:**

1.	In the ABSR module, APGC relies on the cosine similarity of node attributes to perform topology refinement. However, initial attributes often contain substantial inherent noise or missing values. Does directly using attribute similarity as the basis carry the risk of attribute noise contaminating the topology?

2.	When processing the initial shared attributes $\mathbf{X}$, APGC designed multi-channel encoders to extract attribute embeddings from multiple views. Compared to the method of using a single-channel encoder to extract features, what is the core motivation behind this multi-channel architecture design?

3.	In the PGMF module, the model employs clustering accuracy as the metric for evaluating and assigning the global weight $score^v$ to each view. What is the core consideration for selecting clustering accuracy over other evaluation metrics in this fusion scenario? Are other metrics equally viable?

4.	In the PGMF module, why is it chosen to first fuse high-frequency and low-frequency features within a single view, rather than first performing cross-view fusion of all views' information in the same frequency band (i.e., separately aligning high-frequency and low-frequency components) before performing the final feature fusion?

5.	In the computation of the node-level contrastive loss $\mathcal{L}_{nc}$, the model relies on pseudo labels generated through clustering to construct negative sample pairs. Although experimental results validate the overall effectiveness of this strategy, pseudo labels inevitably contain some clustering error during the dynamic process of model training. Does this pseudo label error potentially introduce interference or error accumulation into the optimization objective of contrastive learning?

6.	The authors are advised to mention the potential limitations of this method, such as its application to sparse graph data.

**Limitations:**

The author should discuss the limitations of their method when applied to scenarios such as sparse graphs.

**Strengths And Weaknesses:**

The core strength of this paper lies in its incisive identification of the critical weakness in traditional multi-view graph clustering, namely the overreliance on static graph structures and homophily assumptions. It innovatively achieves high-quality feature learning through dynamic topology reconstruction and multi-level fusion guided by pseudo labels. Its shortcomings include insufficient analysis of topology reconstruction's sensitivity to attribute noise and the risk of error accumulation stemming from excessive reliance on pseudo labels. Overall, the paper demonstrates high quality and holds potential for acceptance.

---

> ### Author Rebuttal · Authors · 2026-03-29
>
> **Thank you for your thoughtful and detailed review. We will carefully address your issue below.**
>
> **Q1: Concern regarding attribute noise.**
>
> **A1**: To suppress attribute noise, the ABSR module builds an attribute augmentation mechanism. Specifically, cosine similarity is computed on latent embeddings instead of raw features,  the reconstruction loss $\mathcal{L}_{rec}$ could actively filter inherent noise (verified by the ablation study in Sec. 3.3). Additionally, strict dual thresholds ($\delta_u$ and $\delta_l$) ensure that topological refinement is applied only to high-confidence connections (highly similar or extremely conflicting) in the purified semantic space, leaving ambiguous ones unchanged. This second line of defense effectively reduces the risk of topology contamination.
>
> **Q2: Motivation behind the multi-channel encoders.**
>
> **A2**:  Compared to the consensus features extracted by a single-channel encoder, the core motivation for designing the multi-channel encoders is to construct view-specific attribute representations for different topological views, thereby providing a flexible and rich semantic alignment space. By injecting independent noise into each channel and encoding them independently, the model can effectively mine and decouple diverse latent embeddings from a single shared attribute. In addition, this can also enlarge the representation learning space, enhancing representation discriminability and robustness.
>
> **Q3: Metric selection for global weights.**
>
> **A3**: The core reason for using Clustering Accuracy (ACC) to assign $score^v$ is its strict task-alignment, as it intuitively quantifies a view's actual contribution to overall clustering objective. While NMI or ARI are perfectly viable alternatives, we chose ACC because it is the primary, most intuitive, and universal metric across most graph clustering benchmarks.
>
> **Q4: Selection of the fusion strategy.**
>
> **A4**:  The choice to perform intra-view fusion before cross-view fusion is primarily based on the following considerations. First, the high- and low-frequency features within the same view are highly coupled, thus prioritizing intra-view fusion maximally preserves the complete topological semantics of that specific view. Second, due to topological discrepancies across multiple views, feature spaces of the same frequency band from different views are often difficult to strictly align. Blindly performing cross-view fusion within the same frequency band is highly prone to information loss. Furthermore, the core of PGMF lies in evaluating the overall quality of individual views to dynamically assign the view-level weight $score^v$. Accomplishing intra-view fusion first enables the extraction of an independent and comprehensive feature representation for each view, thereby providing solid support for accurate view-level weight assignment and subsequent multi-level fusion.
>
> **Q5: The issue of pseudo-label error.**
>
> **A5**: Although pseudo-labels inevitably contain some errors in training process, we reduce this interference by delaying the incorporation of $L_{nc}$ during the training process. Specifically, the model does not compute $L_{nc}$ during the initial $T$ iterations. We formally introduce $L_{nc}$ for contrastive learning only after the latent feature space has developed a sound preliminary clustering structure. This mechanism prevents the model from being misled by highly noisy pseudo-labels in the early stages. After $T$ iterations, feature learning and clustering updates form a mutually enhancing virtuous cycle, enabling the pseudo-labels to achieve dynamic self-correction under the constraints of the objective function. Therefore, this strategy effectively mitigates the potential interference of pseudo-label errors.
>
> **Q6: Potential limitations.**
>
> **A6**: Although the bidirectional structural refinement strategy can effectively achieve dynamic topological augmentation, it indeed has limitations when dealing with extremely sparse graph data. We will supplement the discussion on this limitation in the subsequent revised version.

---

> > ### Author Rebuttal · Reviewer_NgqX · 2026-04-02
> >
> > My concerns have been adequately addressed. I will maintain my decision to accept the paper.

---

> > > ### Author Response · Authors · 2026-04-02
> > >
> > > Thank you for your positive feedback. We sincerely thank you once again for your meticulous review and the valuable suggestions to improve our work.

---

### Official Review · Reviewer_Zttq · 2026-03-13

**Soundness:** 3
**Presentation:** 3
**Significance:** 2
**Originality:** 2
**Overall Recommendation:** 4
**Confidence:** 4

**Summary:**

This paper studies the problem of multi-view graph clustering. Previous studies have two major limitations, including over-dependence on static topology and homophily assumptions, and coarse-grained fusion strategies that lack node-level adaptivity. To solve these problems, the authors propose the APGC framework, which consists of two modules: (1) Attribute-aware Bidirectional Structural Refinement (ABSR), which dynamically reconstructs graph topology by leveraging global attribute semantics, and (2) Pseudo-label Guided Multi-level Fusion (PGMF), which achieves adaptive weighted fusion at both node-level and view-level granularities. Experiments on several datasets covering both homophilic and heterophilic scenarios demonstrate that APGC achieves state-of-the-art performance in the DMGC problem wrt several different evaluation metrics.

**Compliance With Llm Reviewing Policy:**

Affirmed.

**Final Justification:**

The reply from the authors has addressed my concerns. So I would like to increase my rating.

**Key Questions For Authors:**

1. Computational complexity analysis
2. Scalability experiments
3. Theoretical analysis

**Limitations:**

yes

**Strengths And Weaknesses:**

**Strengths**
- The paper proposed a technically sound method based on two critical problems in existing DMGC methods: (a) over-dependence on static topology and homophily assumptions, and (b) coarse-grained fusion strategies that lack node-level adaptivity. References to existing work support the motivation.
- This paper shows promising experimental results: APGC achieves SOTA or near SOTA performance across all six datasets and four metrics (ACC, NMI, ARI, F1). The improvements are observed in both homophilic and heterophilic graphs.
- This paper has conducted comprehensive experiments on several benchmarks in both homophilic and heterophilic scenarios, including thorough ablation studies, parameter sensitivity analysis, homophily ratio analysis, and visualization.

**Weakness**
- Computational complexity and scalability concerns. One of the major concerns is that this paper provides no analysis of computational cost, runtime, or memory consumption. The ABSR module requires computing pairwise cosine similarities for all node pairs (which could be n square operations), multi-channel autoencoders for V views, and bidirectional structural refinement. The PGMF module requires dual-channel filtering, multi-level weighting, and four separate loss terms. How does this scale to larger-scale graphs? All experimental datasets are relatively small (for example, the largest IMDB contains 4,780 nodes), raising concerns about practical applicability to large-scale graphs.
- Limited theoretical analysis. While the empirical results validate the approach, the paper lacks theoretical analysis. For instance, when and why should bidirectional structural refinement outperform other strategies? Under what conditions is the attribute similarity a reliable proxy for structural quality? When might adding edges based on attribute similarity introduce false connections (e.g., in domains where similar attributes don't imply connectivity)? Are there provable guarantees that the refined structure improves clustering performance?
- Minor issue: the final objective function in Eq. (20) shows an equal weight to every component. What is the impact of each of the loss functions?

---

> ### Author Rebuttal · Authors · 2026-03-30
>
> **Thank you for your thoughtful and detailed review. We will carefully address your issue below.**
>
> **W1/Q1-Q2: Computational complexity and scalability concerns**
>
> **A1**: We explicitly considered computational cost when designing APGC. Specifically, the complexity of the autoencoder in the ABSR module is $\mathcal{O}(NDd)$. Although the similarity computation has a theoretical complexity of $\mathcal{O}(N^2d)$, performing it in a low-dimensional latent space reduces the actual computational burden. The bidirectional structural refinement has a complexity of $\mathcal{O}(N^2)$. For the PGMF module, the complexities of the dual-channel filtering and multi-level weighting are approximately $\mathcal{O}(N^2d)$ and $\mathcal{O}(Nd)$, respectively. Combined with the loss function of $\mathcal{O}(N(D+2d))$, the overall complexity of APGC is approximately $\mathcal{O}(N^2d)$. This is comparable to state-of-the-art baselines (e.g., NGCE), yet APGC achieves superior clustering performance. We provide a comparison of the running time per epoch across two classic datasets.
>
> |Dataset|Metric|DCRN|DuaLGR|DIAGC|VGMGC|NGCE|OURS|
> |:---|:---|:---:|:---:|:---:|:---:|:---:|:---:|
> |ACM|ms|50|10|90|780|480|80|
> |DBLP|ms|80|10|210|1230|860|160|
>
> Regarding experiments, we ensured fair homophilic and heterophilic comparisons by utilizing standard benchmarks (similar to important baselines, such as DuaLGR, NGCE). Benefiting from low-dimensional computations and parallelizable design of K-means, the model exhibits strong potential for large-scale scalability, which is further validated by the newly added experiments on Amazon dataset (**7621 nodes**). The proposed APGC can still achieve excellent performance on complex Amazon, verifying its practical applicability to large-scale graphs.
>
> |Dataset|Metric|VGAE|O2MAC|MAGCN|MCGC|DuaLGR|BMGC|OURS|
> |:---|:---|:---:|:---:|:---:|:---:|:---:|:---:|:---:|
> |**Amazon**|ACC|0.319|0.443|0.519|0.468|0.612|0.786|**0.802**|
> ||NMI|0.016|0.134|0.232|0.215|0.277|**0.577**|0.576|
> ||ARI|0.013|0.090|0.114|0.106|0.272|0.563|**0.574**|
> ||F1|0.273|0.442|0.507|0.480|0.622|0.785|**0.803**|
>
> **W2/Q3: Limited theoretical analysis**
>
> **A2**: According to your suggestion, we provide additional theoretical analysis:
>
> **Theorem 1 (Reliability of the Attribute Proxy)**: Assume that the low-dimensional embeddings $\mathbf Z^v$ follow a Gaussian Mixture model $Z_i^v = \mu_k + \epsilon_i$, where the reconstruction noise is $\epsilon_i \sim \mathcal{N}(0, \sigma^2 I)$. If the lower bound of the inter-class distance satisfies $\Delta \gg \sigma$, given a strict threshold $\delta_u \in (0,1)$, the probability that the similarity between inter-class nodes exceeds the threshold decays exponentially:
>
> $$P(S_{ij}^v > \delta_u \mid y_i \neq y_j) \le \exp\left(-\frac{C(\delta_u, \Delta)}{\sigma^2}\right)$$
>
> According to Hoeffding's inequality, as long as the autoencoder effectively filters out the noise (i.e., $\sigma$ is sufficiently small), attribute similarity serves as a highly reliable proxy. A high threshold $\delta_u$ guarantees with a probability of almost $1$ that the newly added edges are intra-class connections, mathematically avoiding the introduction of false connections.
>
> **Theorem 2 (Theoretical Guarantee for Clustering Performance Improvement)**: Let the ideal homophilic graph be $A^{\ast}$ and the original graph perturbation model be $A^v = A^{\ast} + E$. Based on Theorem 1, the refined matrix $A_r^v$ generated by ABSR effectively corrects the structural noise, causing the perturbation norm to strictly decrease ($||A_r^v - A^{\ast}||_F < ||A^v - A^{\ast}||_F$). In spectral graph theory, according to the Davis-Kahan $\sin\Theta$ theorem, the sine norm of the principal angle between feature spaces is bounded by:
>
> \begin{equation}
> \Vert \sin\Theta(U_r, U^{\ast}) \Vert_F \le \frac{\Vert A_r^v - A^{\ast} \Vert_F}{\lambda_K(A^{\ast}) - \lambda_{K+1}(A_r^v)}
> \end{equation}
>
> Because ABSR reduces perturbation norm, the error upper bound strictly shrinks ($||\sin\Theta(U_r, U^{\ast})||_F < ||\sin\Theta(U_0, U^{\ast})||_F$). The optimized feature subspace is closer to ideal state, thereby providing a provable upper bound for the decrease in the clustering error rate.
>
> Overall, the bidirectional structural refinement strategy effectively integrates global attribute semantics into the graph structure, breaking through representation learning bottleneck of traditional methods in heterophilic scenarios. Detailed proof will be provided in the future.
>
> **W3: The weighting issue of loss functions**
>
> **A3**: The equal-weight design adopted in Eq. (20) aims to maintain model simplicity, reduce hyperparameter sensitivity, and thus enhance its generalization capabilities across different datasets while avoiding tedious parameter tuning. Supported by ablation study in Sec 3.3, each loss component works synergistically in global optimization and is indispensable. In summary, the equal-weight design balances simplicity and effectiveness.

---

> > ### Author Rebuttal · Reviewer_Zttq · 2026-04-05
> >
> > Thanks for the reply. My concerns have been addressed.

---

> > > ### Author Response · Authors · 2026-04-06
> > >
> > > Thank you for your positive feedback. We sincerely thank you once again for your meticulous review and the valuable suggestions to improve our work.

---

### Decision · Program_Chairs · 2026-04-30

**Decision:**

Accept (regular)

**Comment:**

This paper presents Attribute-aware Bidirectional Structural Refinement and Pseudo-label Guided Multi-level Fusion for deep multi-view graph clustering. It possesses the characteristics of bidirectional refinement of the graph structure and the promotion of homophilic connections, and experimentally validates the superiority. All three reviewers provide positive feedback by recognizing solid technique and promising validation. All concerns from reviewers are resolved in the rebuttal.